# Interhemispherically dynamic representation of an eye movement-related activity in mouse frontal cortex

**Takashi R Sato[1,2,3,4†]\*, Takahide Itokazu[2,5†], Hironobu Osaki[2,6†], Makoto Ohtake[1,7], Tetsuya Yamamoto[7], Kazuhiro Sohya[3], Takakuni Maki[8], Tatsuo K Sato[4,9,10]\***

[1]Department of Neuroscience, Medical University of South Carolina, Charleston, United States; [2]Center for Integrative Neuroscience, University of Tübingen, Tübingen, Germany; [3]Department of Mental Disorder Research, National Institute of Neuroscience, National Center of Neurology and Psychiatry, Tokyo, Japan; [4]JST, PRESTO, Kawaguchi, Japan; [5]Department of Neuro-Medical Science, Osaka University, Osaka, Japan; [6]Department of Physiology, Tokyo Women's Medical University, Tokyo, Japan; [7]Department of Neurosurgery, Yokohama City University Graduate School of Medicine, Yokohama, Japan; [8]Department of Neurology, Kyoto University Graduate School of Medicine, Kyoto, Japan; [9]Department of Physiology, Monash University, Clayton, Australia; [10]Neuroscience Program, Biomedicine Discovery Institute, Monash University, Clayton, Australia

**Abstract** Cortical plasticity is fundamental to motor recovery following cortical perturbation. However, it is still unclear how this plasticity is induced at a functional circuit level. Here, we investigated motor recovery and underlying neural plasticity upon optogenetic suppression of a cortical area for eye movement. Using a visually-guided eye movement task in mice, we suppressed a portion of the secondary motor cortex (MOs) that encodes contraversive eye movement. Optogenetic unilateral suppression severely impaired contraversive movement on the first day. However, on subsequent days the suppression became inefficient and capability for the movement was restored. Longitudinal two-photon calcium imaging revealed that the regained capability was accompanied by an increased number of neurons encoding for ipsiversive movement in the unsuppressed contralateral MOs. Additional suppression of the contralateral MOs impaired the recovered movement again, indicating a compensatory mechanism. Our findings demonstrate that repeated optogenetic suppression leads to functional recovery mediated by the contralateral hemisphere.

**\*For correspondence:**
satot@musc.edu (TRS);
tatsuo.sato@monash.edu (TKS)

†These authors contributed equally to this work

**Competing interests:** The authors declare that no competing interests exist.

## Introduction

Neural plasticity in motor cortex is critical not only for motor learning (*Peters et al., 2017*), but also for motor recovery following cortical damage (*Nudo, 2013*). Motor plasticity has been traditionally investigated using motor cortex in higher animals (*Travis and Woolsey, 1956*). However, recent technical advancements have rendered rodent cortex a fruitful model to study circuit mechanisms in motor learning (*Makino et al., 2016*), motor deficits (*Ebbesen and Brecht, 2017*) and motor recovery (*Murphy and Corbett, 2009*). For example, previous studies in rodents investigated the cellular and molecular mechanisms that underlie compensatory pathophysiological changes in the cortical network during stroke recovery (*Alia et al., 2017*; *Fawcett, 2015*; *Li et al., 2010*; *Schwab and Strittmatter, 2014*). Such changes include modification of extracellular matrix structures (*Fawcett, 2015*) and increased neurotropic factors for angiogenesis, neurogenesis, and synaptic plasticity

(*Berretta et al., 2014*). However, it is not clear whether these molecular processes are necessary to recover impaired movements or whether motor recovery can be mediated simply by mechanisms similar to motor learning inherent to the physiological circuits.

Recovery from motor deficits relies on the intact brain regions including a hemisphere contralateral to cortical lesions. Indeed, the contralateral hemisphere has been a target for motor rehabilitation in humans (*Buetefisch, 2015*). To investigate the role of the contralateral hemisphere in motor recovery, one of ideal models is neural circuits underlying eye movement. Eye movement, like binocularly coupled saccade, shows a simple but robust motor output, and its direction is represented mainly in the contralateral frontal cortex. Consistent with this representation, a unilateral lesion in primate frontal cortex disrupts saccades toward the contralateral side (i.e., contraversive saccades) (*Crowne et al., 1981*; *Pierrot-Deseilligny et al., 2002*; *van der Steen et al., 1986*). However, such deficits can ease over time (*Crowne et al., 1981*; *van der Steen et al., 1986*), even after the removal of an entire cortical hemisphere (*Bruell and Volk, 1956*; *Estañol et al., 1980*; *Herter and Guitton, 2004*; *Perenin and Jeannerod, 1978*; *Sharpe et al., 1979*; *Troost et al., 1972*; *Tusa et al., 1986*). The recovery of the contraversive saccade in primate implies compensatory neural plasticity, potentially involving the contralateral hemisphere.

To investigate the neural basis for motor recovery in eye movement, we optogenetically suppressed the unilateral mouse frontal cortex during a visually-guided eye movement task (*Itokazu et al., 2018*). Using this task, we previously demonstrated that a small portion of the secondary motor cortex (MOs) in the frontal cortex controls contraversive eye movements, optogenetic suppression of MOs impairs contraversive eye movements, and MOs neurons preferentially encode contraversive eye movements. Here, we found that the suppressed eye movement can be recovered over time due to plasticity of the contralateral MOs. We propose that the neural representation of motor output is highly plastic even without pathophysiological events. Such plasticity could represent the neural basis of motor recovery.

## Results

### The MOs primarily encodes contraversive eye movement condition

To establish a circuit basis of the neural plasticity underlying motor recovery, we investigated the neural representation of eye movements during a visually-guided eye movement task that we previously developed (*Itokazu et al., 2018*). In this task, a head-fixed mouse first directs its left eye toward a central fixation light-emitting diode (LED), and then moves its left eye in the direction of a target LED that is illuminated on either the nasal or temporal side (*Figure 1A*, see Materials and methods). After the target LED illumination, the mice were required to perform eye movements. After several weeks of training, the mice became capable of performing eye shifts in the appropriate direction within 10 s to obtain reward with 84.7 ± 2.5% success (n = 10 mice, See Materials and methods). As we previously reported (*Itokazu et al., 2018*), even though the task required only left eye movement, movements of both eyes were coupled (*Figure 1B*).

Using our task, we investigated the movement direction preference of neurons in the secondary motor cortex (MOs), a motor area for eye movement (*Itokazu et al., 2018*). We monitored MOs neural activity in vivo by imaging the virally expressed genetically encoded calcium indicator GCaMP6m (see Materials and methods, *Figure 1C and D*). Among the imaged neurons, 47.0% showed a significant increase in activity just before eye movement onset (n = 463/985 neurons in ten mice). This activity – which we refer to as eye movement-related activity – is considered a mixture of motor command and visual activity (*Itokazu et al., 2018*) due to the short separation between visual cue onset and motor onset. To evaluate the contraversive or ipsiversive preference of the eye movement-related activity, we computed the difference in ΔF/F increase (see Materials and methods). As expected based on our previous report (*Itokazu et al., 2018*), the distribution of selectivity was significantly skewed to the positive value (*Figure 1E and F*, Wilcoxon signed-rank test, $p<10^{-10}$), indicating the preference for the contraversive condition as a population. In addition, more individual neurons showed significantly higher activity in the contraversive eye movement condition (24.5% neurons, contraversive selective) than the ipsiversive one (7.5%, ipsiversive selective), confirming that the MOs mainly encodes contraversive eye movement condition.

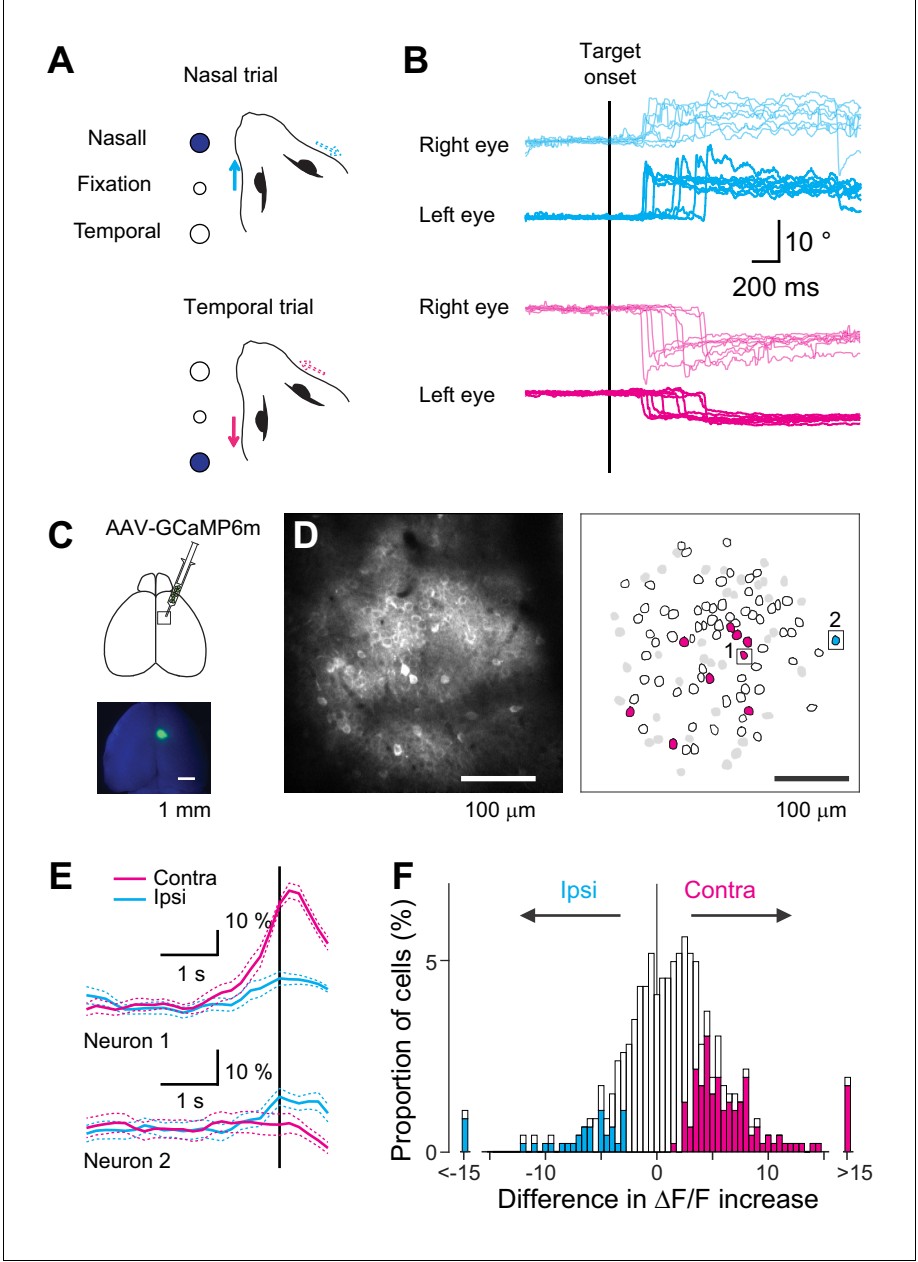

**Figure 1.** Neural representation in the MOs during a visually-guided eye movement task. (**A**) Experimental design of a visually-guided eye movement task. After the mice fixated the central LED, nasal or temporal target LED was turned on, instructing the mice to shift their left eye toward the target. (**B**) Example traces of eye position recorded during one behavioral session. Traces are aligned to the target onset. Magenta traces: trials with temporal target (n = 8 trials). Cyan traces: trials with nasal target (n = 8 trials). (**C**) GCaMP6m was virally expressed in the MOs in the right hemisphere for two-photon calcium imaging. (**D**) Representative image and corresponding ROIs for neurons labeled with GCaMP6m. Magenta polygons: neurons exhibiting higher activity in the contraversive than ipsiversive condition just before eye movement onset. Cyan polygons: neurons exhibiting higher activity in the ipsiversive condition. White polygons: neurons showing significant movement-related activity. Gray polygons: neurons showing no significant movement-related activity. Black squares: example neurons shown in E. (**E**) Fluorescence changes for the two neurons shown in D. Average fluorescence changes for contraversive trials (magenta) and ipsiversive (cyan). Vertical line indicates eye movement onset. (**F**) Distribution of contraversive/ipsiversive difference in ΔF/F increase for neurons showing significant fluorescence change (n = 463). Magenta bars: neurons significantly selective for the contraversive condition (n = 114). Cyan bars: neurons significantly selective for the ipsiversive condition (n = 35).

*Figure 1 continued on next page*

*Figure 1 continued*
The online version of this article includes the following source data and source code for figure 1:
**Source data 1.** Contains numerical data plotted in *Figure 1F*.
**Source code 1.** Displays the distribution of difference in DF/F increase.

## Unilateral MOs suppression impairs contraversive eye movement only on the first day

Previous behavioral lesion studies in primates suggest that deficits in eye movement can be recovered. We reproduced this behavioral result for the first time in rodents by unilateral optogenetic suppression across several days. Previously, we demonstrated that optogenetic suppression of the MOs during the eye movement task impeded contraversive eye movement severely and had non-significant effects on ipsiversive movements (*Itokazu et al., 2018*). Therefore, in this study, except for the two control mice (n = 2 mice) that received both ipsiversive and contraversive suppression, we tracked the impact of unilateral suppression only on contraversive eye movement (n = 8 mice, Materials and methods). We achieved optogenetic suppression in randomly interleaved trials by locally activating parvalbumin (PV) interneurons that expressed channelrhodopsin-2 (ChR2) via viral transduction, and maintained the suppression for 1 s after target LED onset (*Figure 2A*).

The unilateral optogenetic suppression turned out to be ineffective on the subsequent days. On the first day of the optogenetic experiment, although the mice could produce contraversive eye movements within a reaction time of 10 s (with suppression: proportion of trials with a reaction time less than 10 s, 75.6%, reaction time, 1.76 ± 1.47 s, n = 127; without suppression: 85.0%, 1.09 ± 1.72 s, n = 193), they could not during the 1 s suppression period (*Figure 2A,B and D*; trials with optogenetic suppression: proportion of trials with a reaction time less than 1 s, 7.1%, n = 127; trials without suppression: 40.4%, n = 193 trials; $p<10^{-10}$, Pearson's chi-square test, see also *Itokazu et al. (2018)*. However, after several days (5.9 ± 1.0 days, n = 8 mice) of optogenetic suppression, the mice regained their contraversive eye movements even during the suppression (*Figure 2C and E*; trials with optogenetic suppression: proportion of trials with a reaction time less than 10 s, 90.0%, less than 1 s, 44.5%, reaction time, 1.06 ± 1.00 s, n = 220 trials; trials without suppression: less than 10 s, 90.7%, less than 1 s, 52.7%, 0.78 ± 1.40 s, n = 150 trials; comparison between with and without suppression, p>0.12, Pearson's chi-square test; comparison between suppression on the first day and one on the last day, $p<10^{-12}$, Pearson's chi-square test), consistent with the behavioral recovery observed in primate lesion studies (*van der Steen et al., 1986*). The recovery from the initial functional deficit implies that this eye movement might be newly encoded in other brain regions.

## Unilateral MOs suppression alters the direction preference of MOs neurons in the contralateral hemisphere

We hypothesized that the MOs in the unsuppressed contralateral hemisphere may be responsible for the recovered movement. If this were the case, the MOs neurons in the unsuppressed contralateral hemisphere might change how they encode movements, perhaps by encoding a de novo preference for recovered eye movement the direction of which is ipsiversive. To investigate the directional preference of the MOs neurons in the unsuppressed hemisphere, we injected additional virus to express GCaMP6m for in vivo two-photon calcium imaging (e.g., suppressing the right hemisphere and imaging the left hemisphere in *Figure 3A*). A few weeks later, following sufficient viral expression, we mapped the response patterns of individual neurons in the MOs while the mouse performed the task ('before' condition, *Figure 3B and D*). Then, we performed optogenetic suppression for contraversive eye movement trials. The contraversive movement was severely impaired on the first day, consistent with *Figure 2B*. However, the movement was eventually recovered (6.8 ± 1.0 days, n = 6 mice; as in *Figure 2C*). At this point, we performed two-photon imaging again to examine the response patterns of individual neurons without optogenetic MOs suppression ('after' condition, *Figure 3A*). We found neurons that showed a new preference for the ipsiversive movement and those that no longer preferred the contraversive movement (*Figure 3D and E*). As a population, many neurons showed an increased preference for the ipsiversive direction or a reduced preference for the contraversive direction (data points below the scatter plot unity line, *Figure 3F*,

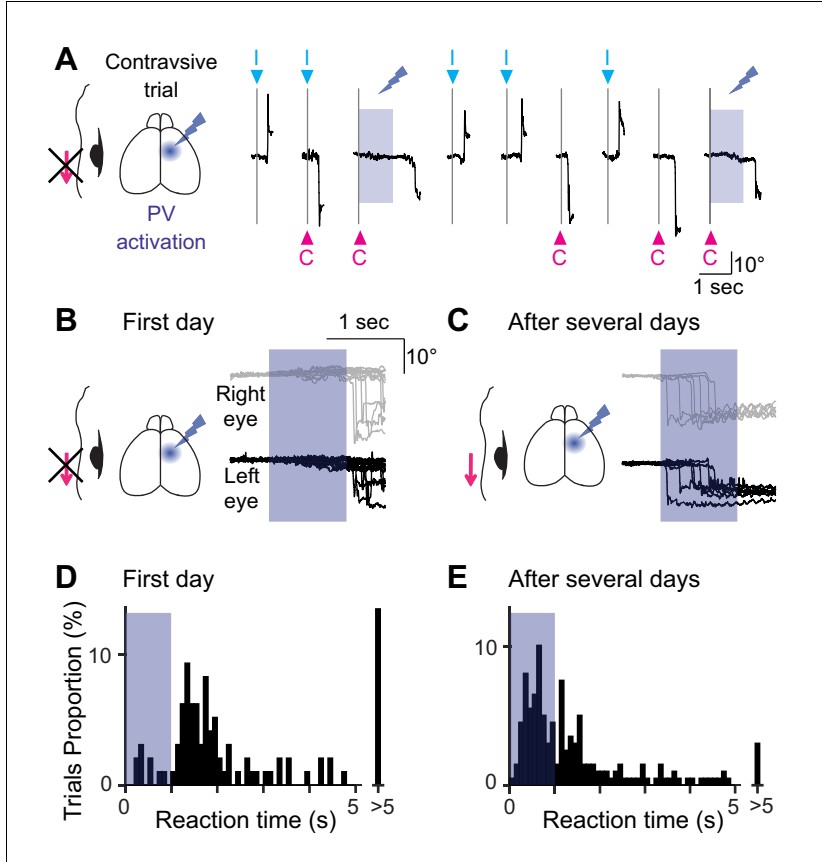

**Figure 2.** Optogenetic suppression of unilateral MOs during the visually-guided eye movement task for multiple days. (**A**) The experimental design and example traces of eye position from one animal. Neuronal activity of the unilateral MOs was optogenetically suppressed during the eye movement task by activating PV+ interneurons. Optogenetic suppression was induced just after visual cues (blue shaded period) only for 1 s. When the blue light was illuminated, the mice could not produce temporal eye movements for the contraversive direction. (**B, C**) Effect of the unilateral optogenetic suppression during the task. Example traces from one animal. The suppression severely impaired the contraversive eye movements on the first day (**B**) but not after 4 days (**C**). Traces of the bilateral eyes are shown. The blue shades indicate the optogenetic suppression period. (**D, E**) Distribution of reaction time for the first day (1.76 ± 1.47 s, 96 out of127 trials, eight mice) and after 5.9 ± 1.0 days (1.06 ± 1.00 s, 198 out of 220 trials, eight mice). The reaction times were significantly different (p<10$^{-9}$, Mann-Whitney U test). The online version of this article includes the following source data and source code for figure 2:

**Source data 1.** Contains numerical data plotted in *Figure 2D,E*.
**Source code 1.** Displays distributions of reaction time.

n = 415 eye movement-related neurons in six mice), especially for neurons that showed direction-selectivity in the 'before' or 'after' conditions (n = 190 out of 415, black circles). Consistently, the distribution for the change in selectivity index (see Materials and methods) was biased toward the ipsiversive condition for direction-selective neurons (p<10$^{-8}$, Wilcoxon signed-rank test, n = 190, black bars, top-right inset in *Figure 3F*), and for all neurons that showed a significant movement-related activity (p<10$^{-6}$, n = 415, black plus white bars). These changes were not simply caused by the training over several days. In control mice that expressed only GFP and thereby received no optogenetic suppression, additional training did not cause an increased ipsiversive preference, but rather led to a tendency for an increased contraversive preference (p=0.13, n = 123 for all direction-selective neurons; p<0.002, n = 282 for all neurons that showed significant motor-related activity, *Figure 3—figure supplement 1*). Therefore, optogenetic suppression caused an increase in neurons selective for ipsiversive direction and a decrease in those selective for contraversive direction, accompanied by a learned capacity to make ipsiversive eye movements.

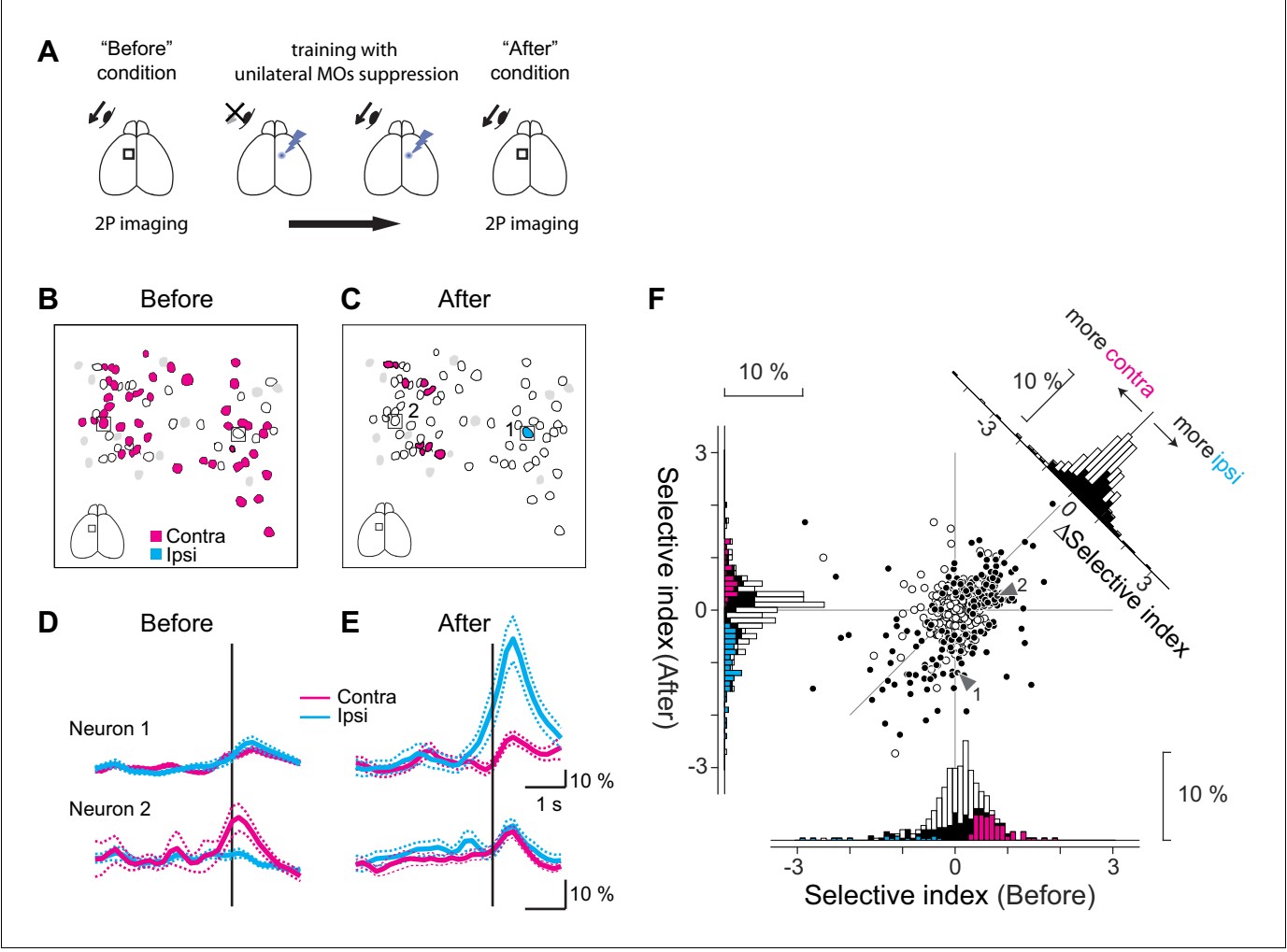

**Figure 3.** Repetitive suppression of unilateral MOs induces compensatory changes in neural encoding in the contralateral MOs. (**A**) Experimental design. When the mice learned the visually-guided eye movement task, we performed two photon calcium imaging to investigate directional preference of the MOs neurons in one hemisphere ('Before' condition). Then optogenetic suppression was applied to the MOs in the contralateral hemisphere. While the mice were trained for several days, they regained the ability to make contraversive eye movements. We again performed two-photon calcium imaging ('After' condition) to obtain the neuronal response for the same neurons imaged in the 'Before' condition. (**B, C**) Training-induced changes in direction selectivity of the MOs neurons in a representative imaging session. In the 'Before' condition (**B**), many neurons preferred the contraversive direction (magenta polygons), but in the 'After' condition (**C**), these neurons got sparser, and one neuron selective for ipsiversive direction (cyan polygon) showed up. White polygons indicate movement-related cells, and gray polygons non-significant cells. Black squares: example neurons shown in D, E. (**D, E**) Examples of two neurons (indicated in B, C) where direction selectivity changed between the 'Before' and 'After' conditions. Average fluorescence changes for trials with ipsiversive movements (cyan traces: Before, n = 30; After, n = 31 trials) and contraversive movements (magenta traces: Before, n = 35; After, n = 32 trials). (**F**) Training for several days induced difference in selectivity index for MOs neurons (six mice). White dots in the scatter plot (n = 225) are neurons showing significant movement-related activity, and black dots (n = 190) are neurons showing significant selectivity on top of the movement-related activity (total of 415 neurons are shown). The x-axis indicates selectivity index for the Before condition and the y-axis for the After condition. The histograms for the selectivity index are shown along the x-axis (Before) and the y-axis (After). Black bars are neurons showing significant direction preference either 'before' or 'after' conditions (n = 190 neurons). On top of black bars, magenta and cyan bars are overlaid for neurons showing preference for contraversive and ipsiversive directions ('before' condition: contraversive, magenta, n = 78 neurons, ipsiversive, cyan, n = 18 neurons, 'after' condition: contraversive, magenta, n = 40 neurons, ipsiversive, cyan, n = 69 neurons). Change in the selectivity index is shown in a histogram in the top right corner. White bars indicate the movement-related neurons (n = 225) and black bars the direction-selective neurons (n = 190).

The online version of this article includes the following source data, source code and figure supplement(s) for figure 3:

**Source data 1.** Contains numerical data plotted in *Figure 3F*.
**Source code 1.** Displays the scatterplot of Selective Indices.
**Figure supplement 1.** Several days of training induces slight increase in contraversive preference in neural encoding.
**Figure supplement 1—source data 1.** Contains numerical data plotted in *Figure 3—figure supplement 1*.
*Figure 3 continued on next page*

*Figure 3 continued*

**Figure supplement 1—source code 1.** Displays the scatterplot of Selective Indices for control data.

## The MOs in the contralateral hemisphere is responsible for the recovery

Our hypothesis predicts that activity of the MOs contralateral to the optogenetic suppression, including the newly emerged ipsiversive preference, could contribute to the recovery of the impaired eye movement. To investigate the contribution of the contralateral MOs, we suppressed this area after the impaired movement was recovered (*Figure 4*). As it is necessary to suppress the MOs in bilateral hemispheres, we used a transgenic line that expresses ChR2-eYFP in PV-positive interneurons (PV-Cre ×Ai32). As expected from *Figure 2B*, on the first day of unilateral suppression the mice were unable to generate contraversive eye movements during 1 s optogenetic suppression (*Figure 4D*, reaction time with and without suppression, 1.69 ± 1.51 s, n = 35 out of 39 trials, vs

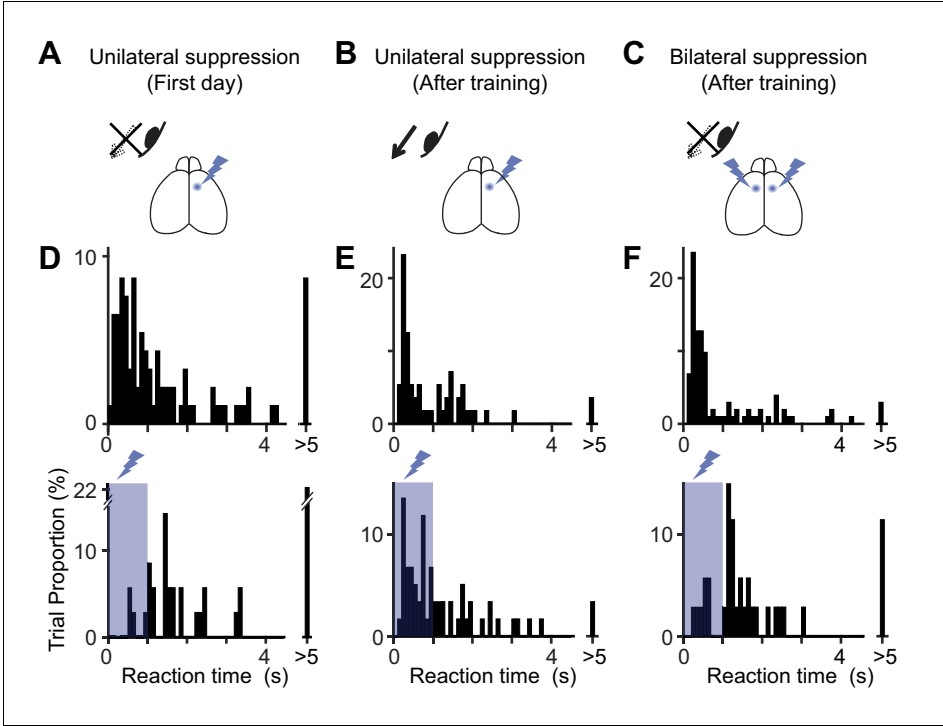

**Figure 4.** Distribution of reaction times after unilateral and bilateral MOs suppression. (**A–C**) Optogenetic suppression design. On the first day of optogenetic suppression (**A**), mice showed difficulty in performing the contraversive eye movements within 1 s of optogenetic suppression. After several days of training (**B**), the mice were able to make contraversive eye movements ('After training'). However, bilateral suppression of the MOs (**C**) induced the difficulty again. (**D–F**) The distribution of reaction times for A-C. Top and bottom histograms are from trials without and with optogenetic suppression. On the first day (**D**), the reaction time is rarely within the 1 s period of optogenetic suppression (10.3% of 39 trials with suppression vs. 52.1% of 96 trials without suppression, $p<10^{-5}$, Pearson's chi-square test, n = 4 mice). After several days of training (**E**), the reaction time could be shorter than 1 s (55.7% of 61 trials with suppression vs. 55.7% of 61 trials without suppression, p=1, Pearson's chi-square test). However, bilateral suppression of the MOs prolonged the reaction time again (18.9% of 37 trials, with suppression vs. 66.7% of 108 trials without suppression, $p<10^{-6}$, Pearson's chi-square test).

The online version of this article includes the following source data and source code for figure 4:

**Source data 1.** Contains numerical data plotted in *Figure 4D*.
**Source data 2.** Contains numerical data plotted in *Figure 4E*.
**Source data 3.** Contains numerical data plotted in *Figure 4F*.
**Source code 1.** Displays the distributions of reaction time.

0.89 ± 1.36 s, n = 92 out of 96 trials, p<0.0002, Mann-Whitney U test, n = 4 mice). Within several days (3.8 ± 0.7 days), the mice regained the capacity to make contraversive movements during unilateral MOs suppression (*Figure 4B and E*, 0.83 ± 0.90 s, n = 59/61 trials vs 0.59 ± 0.90 s, n = 56/61 trials, p>0.07, Mann-Whitney U test). However, additional suppression of the MOs in the contralateral hemisphere caused the same impairment as on the first day (*Figure 4C and F*, 1.27 ± 1.21 s, n = 35/37 trials vs 0.45 ± 0.97 s, n = 102/108 trials, p<10$^{-5}$, Mann-Whitney U test). The reaction time for bilateral suppression was significantly longer than that for post-training unilateral suppression (*Figure 4E and F*, bottom, 1.27 ± 1.21 s vs. 0.83 ± 0.90 s, p<0.02, Mann-Whitney U test), and was similar to the reaction time before training (first day, *Figure 4D* bottom, 1.69 ± 1.51 s, p>0.05, Mann-Whitney U test). The trial proportion with a reaction time less than 1 s was 21.6 ± 8.1% (n = 4 mice) for bilateral suppression, which was significantly lower than that for unilateral suppression (56.0 ± 9.7%, p<0.001, Pearson's chi-square test), but similar to that before training (10.6 ± 5.2% for before training, p>0.25, Pearson's chi-square test). The significant impact of the bilateral suppression demonstrates that the contralateral MOs acquired an essential role in motor recovery after repetitive optogenetic suppression. This essential role, together with new encoding for ipsiversive direction (*Figure 3*), suggests that the dynamic properties of MOs neurons may underlie suppression-induced adaptation.

## Discussion

In this study, we investigated the contribution of the contralateral hemisphere to eye movement recovery following suppression of the unilateral motor cortex. We showed that following unilateral MOs suppression, the contralateral MOs encoded the ipsiversive direction, and played a critical role to compensate for the impaired function of the suppressed hemisphere. To our knowledge, this is the first direct evidence showing dynamic directional encoding of eye movement across hemispheres.

Our finding unveiled a recovery process following unilateral optogenetic suppression. It shares similarity with recovery processes following a unilateral lesion in the motor cortex in human patients and in animal models (reviewed in *Alia et al., 2017*; *Murphy and Corbett, 2009*; *Nudo, 2007*). Cortical lesions are usually induced by events such as cortical resection, aspiration, traumatic brain injuries, and experimental strokes, which severely damage neurons, glial cells, vessels, neighboring axons, and the extracellular matrix including perineuronal nets. This damage initiates highly complex pathophysiologic reactions like inflammation and reactive gliosis, which leads to the upregulation of neurotrophic factors and pro-inflammatory cytokines. All of these reactions could, in principle, be involved in plasticity (*Alia et al., 2017*; *da Silva Meirelles et al., 2017*; *Fawcett, 2015*; *Li et al., 2010*). Therefore, it was not clear whether functional recovery could be achieved without complex pathophysiological events triggered by cortical lesions. Our findings demonstrate that a built-in plastic capability of normal cortical circuits alone could cause motor recovery of eye movement.

Our findings demonstrate that the motor cortex in the contralateral hemisphere underlies the flexibility in the neural representation of motor output. Studies in the forelimb/hindlimb motor cortex suggest that both the perilesional areas and the contralateral unaffected motor cortex play critical roles in motor recovery (*Alia et al., 2017*; *Benowitz and Carmichael, 2010*; *Murphy and Corbett, 2009*). In particular, following a cortical lesion, neurons in the contralateral cortex show increased turnover of dendritic spines and an increased number of dendritic branches. Consistently, following a large unilateral infarct, pharmacological silencing of the contralateral hemisphere can deteriorate recovered forelimb movement, indicating a facilitatory role (*Biernaskie et al., 2004*). However, the role of the contralateral hemisphere still remains controversial (*Alia et al., 2017*; *Hosp and Luft, 2011*; *Mohajerani et al., 2011*), and indeed a study showed that silencing the undamaged contralateral hemisphere by continuous infusion of GABA-A agonist improves recovery performance, implying a suppressive role (*Mansoori et al., 2014*). A prevailing hypothesis that reconciles these findings is that the role of the contralateral hemisphere depends on the size of infarction, particularly on whether remaining intact cortical areas in the ipsi-lesional hemisphere can take over the original functions of the lesioned area (*Buetefisch, 2015*; *Di Pino et al., 2014*). If they can, they will be interfered with by the contralateral hemisphere. If they cannot, the contralateral hemisphere instead can substitute for the functions of the lesioned area. Our findings suggest that

neighboring areas in the hemisphere ipsilateral to the optogenetic suppression, despite potentially exhibiting some forms of plasticity, may not substitute for the role of MOs in our task.

Our findings indicate that the MOs has a latent capability to control ipsiversive eye movements and can learn to turn on these movements. This capability might depend on existing anatomical projections from the MOs to subcortical eye-movement related regions for ipsiversive movements. For instance, the MOs normally projects to the superior colliculus. However, in this case the MOs likely would need to project to the contralateral superior colliculus; these projections are rather sparse (*Oh et al., 2014*, for example, mouse #141603190). Another possibility is the relatively strong projections from the MOs to the contralateral striatum (*Oh et al., 2014*); the striatum has been linked to saccade output as an indirect pathway in primates (*Basso and Sommer, 2011*). Further research is required to determine which anatomical wirings the MOs neurons learn to exploit to achieve motor recovery.

Optogenetic suppression has been a standard technique to investigate the specific functions of neural circuits (*Guo et al., 2014*; *Li et al., 2016*). Our data show that repeated suppression across multiple days can induce changes in the functions of the contralateral hemisphere, leading to the recovery from the initial motor deficits induced by optogenetic suppression. Therefore, our results indicate that the impact of optogenetic suppression needs to be interpreted with caution, and the suppression paradigms need to be carefully designed. Our study highlights the flexibility of cortical circuits that can overcome even short-term reversible manipulation of neural activity.

# Materials and methods

**Key resources table**

| Reagent type (species) or resource | Designation | Source or reference | Identifiers | Additional information |
|---|---|---|---|---|
| Strain, strain background (*Mus musculus*) | PV-Cre, C57Bl/6 | PMID: 15836427 | RRID:IMSR_JAX:008069 | The Jackson Laboratory (#008069) |
| Strain, strain background (*Mus musculus*) | Ai32, C57Bl/6 | PMID: 22446880 | RRID:IMSR_JAX:012569 | The Jackson Laboratory (#012569) |
| Recombinant DNA reagent | AAV2/1-syn-GCaMP6m | PMID: 23868258 | RRID:Addgene_100841 | Upenn Vector Core |
| Recombinant DNA reagent | AAV2/1-EF1α-DIO-hChR2(H134R)-EYFP | http://www.optogenetics.org | RRID:Addgene_ 20298 | Upenn Vector Core |
| Software, algorithm | Matlab | https://www.mathworks.com/products/matlab.html | RRID:SCR_001622 | |

## Animals and surgery

C57BL/6J, PV–Cre, which has the Cre recombinase gene targeted to the *Pvalb* locus (JAX stock #008069), and Ai32 (JAX stock #012569; Rosa-CAG-LSL-ChR2(H134R)-EYFP-WPRE) mouse lines were used in this study. In some experiments, PV–Cre mice were crossed with Ai32 mice, and the resulting mouse line was designated PV–ChR2. For all experiments, male mice of 8 weeks or older in age were used. The mice were group housed in a cage, and experiments were performed during the dark period of the 12 hr light/12 hr dark cycle. Sixteen mice were included in this study. For surgical procedures, mice were anesthetized with 0.1 mg/g ketamine and 0.008 mg/g xylazine (intraperitoneally), and isoflurane was supplemented to maintain the anesthesia. Lidocaine was applied subcutaneously at the incision site. Dexamethasone (2 mg/kg) was administered intraperitoneally after the onset of anesthesia to reduce swelling of the tissue. Lidocaine was applied to the wound margins for topical anesthesia. A custom-built headpost was glued to the skull and then cemented to the animal's head using dental acrylic. A craniotomy (1–2 mm rectangle) was performed over the MOs of one hemisphere (centered 700 µm anterior and 700 µm lateral from the bregma). Then, virus (AAV2/1-syn-GCaMP6m or AAV2/1-EF1α-DIO-hChR2(H134R)-EYFP) was injected at multiple sites (10–20 nL/site; depth, 200–300 µm; 3–5 min/injection), which resulted in virus expression of ~400 µm. The laterality of the hemisphere was randomized across mice. For PV-ChR2 mice, no virus was

injected, but two separate craniotomies were made over the MOs of both hemispheres. Following virus injection, an imaging window consisting of two or three layers of cover glass was implanted. The space between the imaging window and bone was sealed with 1.5% agarose and the window was cemented with dental acrylic (*Komiyama et al., 2010*).

## Behavioral training

Mice were trained on a visually-guided eye movement task that we previously developed (*Itokazu et al., 2018*). In brief, mice were pre-trained to enter a tube to obtain a water reward for ~7 days. Then, the mice were acclimated to the imaging setup (head-fixed condition), and rapid eye movements were encouraged with water reward for 2 days. Following this, the mice were trained to perform a visually-guided eye movement task. In this behavioral paradigm, three blue LEDs (a fixation LED, a nasal target LED, and a temporal target LED; wavelength 470 nm, M470F1, Thorlabs) were used as visual stimuli. During the trial, the brightness of LEDs indicated where the left eye should be located (*Figure 1A*). The light of the fixation LED was first set at 260 μW for 3–4 s, and then increased to 470 μW at the start of the trial. When the left eye was directed to the fixation LED (within ±2.5°), the brightness was increased to 500 μW. The mouse was required to maintain the fixation for 750–1,000 ms. After successful fixation, one of the two target LEDs was turned on (450 μW). At the same time, the brightness of the fixation LED was decreased (260 μW). The mouse needed to shift its gaze in the direction of the target LED within 10 s. The 10 s window for the correct eye movements remains the same for all the trials including those with optogenetic suppression. If the mouse did not make the eye movements within 10 s, the trial was aborted and excluded from the reaction time analysis. Following successful eye movements, the target LED was turned off (70 ms later). After a short delay, a drop of water was provided. The eye shift needed to be a rapid movement (amplitude >5°, speed >0.1°/ms). If the gaze moved out from the fixation window without the correct eye movement, the trial was considered as an error. Incorrect eye movements included rapid eye movements toward the wrong direction, rapid movements with amplitudes smaller than 5°, and slow eye drifts. Each mouse generally performed this task ~50 times per day. The behavior was monitored using a program written in TEMPO (*Itokazu et al., 2018*; *Sato and Schall, 2003*; *Sato et al., 2003*). In this study, we did not force animals to make eye movements at long latency (*Itokazu et al., 2018*) in optogenetic experiments. In pilot experiments, optogenetic suppression discouraged such animals from task engagement.

## Measurement and analysis for eye position

Methods used to measure and analyze eye position have been described in detail previously (*Itokazu et al., 2018*). In brief, the position of the left eye was monitored using a commercial eye-detection package (Eyelink 1000, SR Research, sampled at 500 Hz). This system returns, without saving the original image, analog voltage output that corresponds to the position of the eye. These voltage outputs are fed into our behavior monitoring program. The light source for the camera was a 940 nm LED. To block the infrared light for the right eye, target LEDs, a two-photon laser, an 800 nm long-pass filter (#66–059, Edmund), and a 960 nm short-pass filter (HQ960SP, Chroma) were placed in front of the camera.

In some experiments, the image of the right eye was recorded at 200 Hz using a complementary metal-oxide semiconductor (CMOS) camera (DCC1240M, Thorlabs). The position of the right eye was determined for each frame using a custom-written program in MATLAB (*Itokazu et al., 2018*). The light source for this camera was a 780 nm LED (M780L2, Thorlabs). In front of the camera, a 960 nm short-pass filter (HQ960SP, Chroma) was placed.

## Two-photon imaging

In vivo imaging was performed using a two-photon microscope based on a movable objective microscope system (Sutter) controlled by ScanImage software (*Pologruto et al., 2003*), as previously described (*Itokazu et al., 2018*). The light source was a pulsed Ti:sapphire laser (Chameleon, Coherent), and the laser wavelength was set at 980 nm, which provides a high fluorescent change in GCaMP signal (https://www.janelia.org/lab/harris-lab-apig/research/photophysics/two-photon-fluorescent-probes) and less scattering in the tissue than shorter wavelengths. The objective lens was apochromatic (16×, 0.80 NA, Nikon). Signals were collected using photomultiplier tubes

(Hamamatsu Photonics, H10770PA-40); frame scanning (frame rate ~6 Hz) was used. Images were collected at a depth of 150–300 μm from the dura surface for layer 2/3 neurons. For image analysis, movement artifacts were corrected in two steps: performing a cross correlation-based image alignment (Turboreg) (*Thévenaz et al., 1998*) followed by a line-by-line correction using an algorithm based on a hidden-Markov model (*Dombeck et al., 2007*). Then, the regions of interest (ROIs) containing the neurons were drawn manually, and the pixel values within each ROI were summed to estimate the fluorescence of the individual neuron. $\Delta F/F$ signal was calculated as $(F-F_{baseline})/F_{baseline}$, where $F_{baseline}$ is the baseline fluorescence signal ($30^{th}$ percentile) within each trial.

## Optogenetic suppression

The design of optogenetic stimulation is the same as in our previous report (*Itokazu et al., 2018*). AAV2/1-EF1α-DIO-hChR2(H134R)-EYFP was injected into the MOs of PV–Cre mice (*Lee et al., 2012*; *Olsen et al., 2012*) unilaterally, which resulted in virus expression of ~800 μm. In the control mice (*Figure 3—figure supplement 1*), AAV2/1 CAG-FLEX-EGFP was injected. Bilateral suppression studies employed PV–ChR2 mice, which were produced by crossing PV–Cre to Ai32 (Rosa-CAG-LSL-ChR2(H134R)-EYFP-WPRE) mice. A blue laser (473 nm, CrystaLaser) was coupled to an optic fiber (M15L02, Thorlabs). The output power was manipulated by combining a half-wave plate with a polarizing beamsplitter cube. An optical chopper was used to convert the continuous wave into a 40 Hz pulse (pulse width, 2.5 ms; Edmund optics, 59–894) (*Cardin et al., 2009*). The output of the optic fiber and surface of the cortex were placed on conjugate planes using two convex lenses. A dichroic mirror was placed in the infinity space, and the reflected light was focused onto the sensor of a CMOS camera (Thorlabs). This design enabled monitoring of the precise location of the stimulated site. The blue light was illuminated in 40–60% of the trials, where contralateral eye movement was required. Illumination was made only in trials with contralateral eye movements (n = 8 mice), as optogenetic suppression affects only those movements (*Itokazu et al., 2018*). In two control mice, both contraversive and ipsiversive trials were suppressed, which resulted in impairment only in contraversive trials. The laterality of suppressed hemisphere for contraversive-trial suppression was randomized between mice, although in PV-Cre mice it was based on the virus injection site (which was randomized at the time of virus injection across mice). We illuminated ~30% of the trials on the first day and the day of bilateral suppression; if we used a higher proportion of optogenetic suppression, the mice often stopped the task engagement. The average power of the light at the surface of the cortex was 600–1,200 μW. Optogenetic suppression was applied for multiple days (5.2 ± 0.7 days). During these periods, the mice learned to perform eye movements under optogenetic suppression. In the following session after learning, two-photon imaging was performed without optogenetic suppression in six PV-Cre mice, and bilateral suppression was performed in four PV-ChR2 mice using two separate lasers and optic fibers.

Even for trials with optogenetic suppression, eye movements within 10 s were considered to be a success, and were thereby rewarded with a drop of water. The effect of optogenetic suppression was quantified by comparing the probability of eye movement within 1 s between trials with and without illumination. The probability was also compared between the first and the last day of unilateral optogenetic suppression, and between unilateral and bilateral suppression (Pearson's chi-square test). The distributions of the reaction times were also compared using the Mann-Whitney U test.

This optogenetic stimulation caused no behavioral effects when blue light illumination was targeted to the primary motor cortex (1 mm lateral to the MOs), excluding non-specific effects of blue light illumination (*Itokazu et al., 2018*).

## Image data analysis

For each neuron, movement-related activity was quantified as an increase in $\Delta F/F$ between the frame at baseline (600 ms before the movement onset) and the frame at the time of movement. The neurons were considered to contain significant movement activity if the activity was significant based on the Wilcoxon signed-rank test (p<0.05) in either contraversive or ipsiversive movement conditions. For these neurons, the difference in movement activity between the contraversive or ipsiversive conditions was computed (difference in $\Delta F/F$ increase in *Figure 1F*). The significance of the direction selectivity was determined by the Mann–Whitney U test with p<0.05.

For the animals that were imaged before and after the training with optogenetic suppression, the neurons were considered to contain significant movement activity if the activity was significant (Wilcoxon signed-rank test, $p<0.05$) either before or after the training (white dots/bars; *Figure 3F*). Similarly, the neurons were considered as direction selective if the difference in movement activity was significant (Mann-Whitney U test, $p<0.05$) either before or after the training (black dots/bars; *Figure 3F*). Selectivity index for contraversive/ipsiversive directions was computed as $(R_{contra} - R_{ipsi}) / |R_{contra} + R_{ipsi}|$, where $R_{contra}$ is the response in the contraversive condition and $R_{ipsi}$ is in the ipsiversive. The index could be larger than one or smaller than $-1$ when either of $R_{contra}$ and $R_{ipsi}$ was negative. In addition, we used a common and the larger denominator between 'before' and 'after' conditions, to avoid a small denominator resulting from non-significant $R_{contra}$ and $R_{ipsi}$ (e.g., Neuron two in 'after' condition in *Figure 3E*).

### Experimental design and statistical analysis

Ten mice (PV-Cre and C57BL/6) were used to study the neural representation of eye movements in MOs, eight mice (PV-Cre) to examine the effects of unilateral optogenetic suppression of MOs, six mice (PV-Cre) to monitor the effects of repeated optogenetic suppression on neural representation in MOs, and four mice (PV-ChR2) for bilateral suppression of MOs.

All of the statistical tests were non-parametric, and are indicated in the relevant text or figure legend. Data and traces are show as mean ± s.e.m., except that reaction times are described as median ± m.a.d., because they have highly skewed distributions (*Reddi and Carpenter, 2000*). Our sample sizes are similar to those reported in previous publications (*Itokazu et al., 2018*).

### Code accessibility

The data and computer codes for this study are available as supplementary data.

## Acknowledgements

This work was supported by grants from DFG (SA 2575/2–1, SA 2575/3–1) to TRS and the Japan Science and Technology Agency (PRESTO) to TRS and TKS. TRS and TKS are PRESTO investigators. We thank S Kashiwagi for technical help, D Shimaoka for careful reading our initial manuscript and LL Looger, K Svoboda, J Akerboom, DS Kim, and the GENIE Project at Janelia Farm for making GCaMP6 available. We thank the members of our laboratories for helpful suggestions.

## Additional information

### Funding

| Funder | Grant reference number | Author |
| --- | --- | --- |
| Japan Science and Technology Agency | PRESTO Grant Number JPMJPR1883 | Tatsuo K Sato |
| Japan Science and Technology Agency | PRESTO (Development and Function of Neural Networks) | Takashi R Sato |
| Deutsche Forschungsgemeinschaft | SA 2575/2-1 | Takashi R Sato |
| Deutsche Forschungsgemeinschaft | SA 2575/3-1 | Takashi R Sato |

The funders had no role in study design, data collection and interpretation, or the decision to submit the work for publication.

### Author contributions

Takashi R Sato, Tatsuo K Sato, Conceptualization, Data curation, Formal analysis, Supervision, Funding acquisition, Investigation, Writing—original draft, Project administration; Takahide Itokazu, Hironobu Osaki, Conceptualization, Data curation, Methodology; Makoto Ohtake, Kazuhiro Sohya, Takakuni Maki, Data curation; Tetsuya Yamamoto, Conceptualization

## Author ORCIDs

Takashi R Sato (iD) https://orcid.org/0000-0002-7623-9021
Hironobu Osaki (iD) http://orcid.org/0000-0001-9780-0810
Tatsuo K Sato (iD) https://orcid.org/0000-0002-1279-5125

## Ethics

Animal experimentation: All experimental procedures were approved by the University of Tuebingen (CIN4/11) Medical University of South Carolina (IACUC-2018-00352), and National Center of Neurology and Psychiatry (2014005).

## Decision letter and Author response

Decision letter https://doi.org/10.7554/eLife.50855.sa1
Author response https://doi.org/10.7554/eLife.50855.sa2

# Additional files

## Supplementary files

• Transparent reporting form

## Data availability

The source data are included in the manuscript and supporting files.

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
