## [Decision Letter]

**Acceptance summary:**

This study provides important insights into the plasticity in the secondary motor cortex during recovery from motor deficits as well as important caveats in interpreting experiments using optogenetic inactivation (or more generally, transient inactivation). These results are of great interest to a wide neuroscience audience.

**Decision letter after peer review:**

Thank you for submitting your article "Interhemispherically dynamic representation of an eye movement-related activity in mouse frontal cortex" for consideration by *eLife*. Your article has been reviewed by two peer reviewers, and the evaluation has been overseen by Naoshige Uchida as the Reviewing Editor and Richard Ivry as the Senior Editor. The reviewers have opted to remain anonymous.

The reviewers have discussed the reviews with one another and the Reviewing Editor has drafted this decision to help you prepare a revised submission.

Summary:

In this short report, Sato and colleagues studied the effects of optogenetic inactivation of secondary motor cortex (MOs) on eye movements using a visually-guided eye movement task in mice. The authors show that unilateral inactivation of MOs impaired contraversive eye movements on the first day of manipulation but this effect was greatly reduced after several days of repeated inactivation. This behavioral recovery was accompanied by changes in MOs activity: MOs neurons initially showed significant contraversive preference. However, after repeated unilateral inactivation, the contralateral MOs reduced contraversive preference. Inactivation of bilateral MOs eliminated the recovered movements.

The reviewers found this study to contain very important and impactful results. Optogenetics is very widely used under the assumption that brief stimulations do not cause significant long-lasting changes in the underlying brain mechanisms. The current study clearly refutes that assumption, at least in the context studied here. As such, this study will be of great interest to a wide neuroscience audience.

The reviewers also found the conclusions to be well supported by the data, the manuscript is well-written, and the results are interesting and thought-provoking. However, the reviewers raised some concerns to which we would like to see your response.

Essential revisions:

Can the authors exclude the possibility that the reported plasticity was caused by tissue damage made by the repetitive illumination of blue laser, but not by the reversible inactivation? One possible control experiment would be 2-photon imaging of the suppressed area after the repetitive suppression. The other possibility would be to test whether a repetitive illumination of MOs without the optogenetics will impair the eye movements or not.

Minor points:

1) Although the "success rate" of the animal's performance in the task is one of the key measures in this study, its definition is somewhat ambiguous. Please clarify the definition of a success rate. For example, in the Materials and methods section, "the mouse needed to shift its gaze.… within 10 s." In the subsection “The MOs primarily encodes contraversive eye movement condition”, success rate based on this definition was "84.5% success". On the other hand, in the subsection “Unilateral MOs suppression impairs contraversive eye movement only on the first day”, the success rate was measured as "proportion of trials with a reaction time less than 1 sec". Therefore, the post-suppression% correct was "7.1% " and that of pre-suppression was "40.4%". Is it possible to use the same definition throughout the manuscript for clarity?

2) Based on Figure 2B left, the mouse made correct saccades in the suppression trials after the light was turned off in some trials. This might suggest that mice were rewarded in many of the suppression trials. What was the rate of reward in the suppression trials? Or, what are the success rates in the experimental sessions if a correct saccade within 10 sec was defined as success? Please clarify the reward contingency and definition of "success" in these cases.

3) Please clarify whether the suppression was applied randomly or in a blocked design (i.e. successive trials).

4) Did the authors examine the neural activity in the surrounding region of the suppressed area or in the suppressed area after the repetitive suppression? If these areas did not change their activity, the conclusion of the manuscript can be strengthened.

5) Are there anatomical reports that support the compensation in the contralateral hemisphere?

---

## [Author Response]

Minor points:1) Although the "success rate" of the animal's performance in the task is one of the key measures in this study, its definition is somewhat ambiguous. Please clarify the definition of a success rate. For example, in the Materials and methods section, "the mouse needed to shift its gaze.… within 10 s." In the subsection “The MOs primarily encodes contraversive eye movement condition”, success rate based on this definition was "84.5% success". On the other hand, in the subsection “Unilateral MOs suppression impairs contraversive eye movement only on the first day”, the success rate was measured as "proportion of trials with a reaction time less than 1 sec". Therefore, the post-suppression% correct was "7.1% " and that of pre-suppression was "40.4%". Is it possible to use the same definition throughout the manuscript for clarity?

We thank the reviewers for raising this point, as it allows us to clarify a potential

misunderstanding. When we describe a success rate with a percentage, it indicates the proportion of trials in which the animals can make movements within 10 seconds (leading to a reward). By contrast, when we evaluate the impact of optogenetic suppression, we describe whether animals can make movements during the 1 second suppression period (the duration in which the laser is illuminated). We restricted the laser illumination to 1 second, so that the mouse can still perform the correct eye movements in the remaining 9 seconds, which helps maintain their motivation.

We have now made these definitions clear throughout the manuscript (subsection “The MOs primarily encodes contraversive eye movement condition”, first paragraph, subsection “Unilateral MOs suppression impairs contraversive eye movement only on the first day”, last paragraph, subsection “The MOs in the contralateral hemisphere is responsible for the recovery”, Materials and methods).

2) Based on Figure 2B left, the mouse made correct saccades in the suppression trials after the light was turned off in some trials. This might suggest that mice were rewarded in many of the suppression trials. What was the rate of reward in the suppression trials? Or, what are the success rates in the experimental sessions if a correct saccade within 10 sec was defined as success? Please clarify the reward contingency and definition of "success" in these cases.

We thank the reviewers for pointing this out. Although we had intended to describe such information, we now realize that our description was not clear enough. Our definition of success, and thereby the condition in which animals can get a reward, is the same throughout this study; animals are required to make a correct eye movement within 10 seconds. On the other hand, we evaluate the impact of optogenetic suppression by comparing the number of trials within 1 second with and without laser illumination.

We have modified our descriptions throughout the manuscript to make this clearer (subsection “The MOs primarily encodes contraversive eye movement condition”, first paragraph, subsection “Unilateral MOs suppression impairs contraversive eye movement only on the first day”, last paragraph, subsection “The MOs in the contralateral hemisphere is responsible for the recovery”, Materials and methods).

3) Please clarify whether the suppression was applied randomly or in a blocked design (i.e. successive trials).

We have now clarified that the suppression was randomized (subsection “Unilateral MOs suppression impairs contraversive eye movement only on the first day”, first paragraph).

4) Did the authors examine the neural activity in the surrounding region of the suppressed area or in the suppressed area after the repetitive suppression? If these areas did not change their activity, the conclusion of the manuscript can be strengthened.

We did not examine the neural activity in the suppressed hemisphere, as bilateral optogenetic suppression showed that the activity in the unsuppressed MOs was sufficient to explain the recovery (Figure 4).

Additional plastic changes in other regions are intriguing. Therefore, we now refer to this possibility in the third paragraph of the Discussion.

5) Are there anatomical reports that support the compensation in the contralateral hemisphere?

The compensatory eye movements could be mediated by a number of different anatomical pathways. We have now discussed these possibilities in a new fourth paragraph in the Discussion.